# Enhancing Multi-Modal LLMs Reasoning via Difficulty-Aware Group Normalization

Jinghan Li [1]   Junfeng Fang [2]   Jinda Lu [1]   Yuan Wang [1]   Xiaoyan Guo [1]
Tianyu Zhang [1]   Xiang Wang [1]   Xiangnan He [1]

## Abstract

Reinforcement Learning with Verifiable Rewards (RLVR) and Group Relative Policy Optimization (GRPO) have significantly advanced the reasoning capabilities of large language models. Extending these methods to multimodal settings, however, faces a critical challenge: the instability of *std*-based normalization, which is easily distorted by extreme samples with nearly positive or negative rewards. Unlike pure-text LLMs, multimodal models are particularly sensitive to such distortions, as both perceptual and reasoning errors influence their responses. To address this, we characterize each sample by its **difficulty**, defined through perceptual complexity (measured via visual entropy) and reasoning uncertainty (captured by model confidence). Building on this characterization, we propose **difficulty-aware group normalization (Durian)**, which re-groups samples by difficulty levels and shares the *std* within each group. Our approach preserves GRPO's intra-group distinctions while eliminating sensitivity to extreme cases, yielding significant performance gains across multiple multimodal reasoning benchmarks.

## 1. Introduction

Reinforcement Learning with Verifiable Rewards (RLVR) has enabled significant advances in the reasoning capabilities of both large language models (LLMs) (DeepSeek-AI et al., 2025; Yang et al., 2025a; Lambert et al., 2024) and multi-modal large language models (MLLMs) (Zhang et al., 2025b; Huang et al., 2025a). Within this paradigm, Group

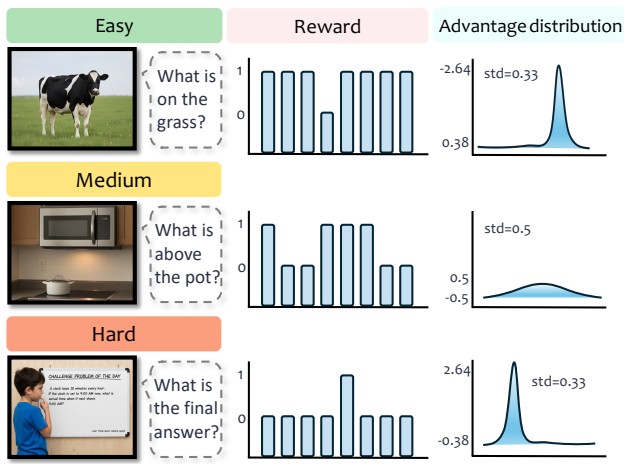

*Figure 1.* The advantage distribution after the normalization of reward varies among samples. Extreme samples like easy and hard ones are amplified after *std*-normalization, whereas medium samples exhibit more balanced advantages.

Relative Policy Optimization (GRPO) (Shao et al., 2024) demonstrates strong performance by applying standard deviation (*std*)-based normalization to rewards within each response group. This *std*-based normalization rescales intra-group distinctions between positive and negative responses, thereby stabilizing training.

Despite these advances, we observe that the *std*-based normalization suffers from a critical limitation: *sensitive to extreme samples — those with response groups that are almost entirely positive or negative*. Specifically, when rewards in a group collapse to near 0 or 1, the resulting low *std* overemphasizes the extreme samples during optimization. Meanwhile, samples with more balanced rewards are neglected, leading to imbalanced optimization. This issue is particularly pronounced in MLLMs, where the complexity of multimodal inputs increases the occurrence of such extreme samples. As illustrated in Figure 1, MLLM responses are jointly influenced by challenges from perceptual complexity and reasoning uncertainty, making them more susceptible to extreme reward distributions.

[1]University of Science and Technology of China [2]National University of Singapore. Correspondence to: Junfeng Fang <fangjf1997@gmail.com>, Xiang Wang <xiangwang1223@gmail.com>.

*Proceedings of the 43rd International Conference on Machine Learning*, Seoul, South Korea. PMLR 306, 2026. Copyright 2026 by the author(s).

A straightforward solution is to remove the *std* term, thereby mitigating the risk of overfitting to extreme samples (Liu et al., 2025b). However, it simultaneously discards the valuable intra-group distinctions, which are essential for effective and stable optimization. Therefore, *the key issue lies not in the std-normalization term itself, but rather in the way groups are constructed*: when group size is small, extreme cases become inevitable. Enlarging group sizes during rollouts could help, but it incurs prohibitive computational costs.

Motivated by this, we propose to account for the challenges of various samples, which we refer to as **Durian: difficulty-aware re-grouping**. We characterize each sample's difficulty from two complementary perspectives: (i) a data-centric view, where the entropy of the image reflects *perceptual difficulty*; and (ii) a model-centric view, where confidence in model responses reflects *reasoning difficulty*. By re-grouping samples according to these difficulty levels and sharing the *std* within each group, our method preserves intra-group distinctions while mitigating sensitivity to extreme cases. Specifically, our difficulty-based re-group strategy is achieved by:

**Perceptual difficulty-based regrouping.** We quantify perceptual difficulty through spectral analysis of image patch covariances, where higher entropy in the resulting eigenvalue distribution indicates greater visual complexity (Grzywacz, 2025). Images with more diverse and complex visual patterns exhibit higher entropy, reflecting greater perceptual difficulty.

**Reasoning difficulty-based regrouping.** Leveraging the insight that token-level log probabilities reflect reasoning confidence (Yu et al., 2025c), we measure reasoning difficulty through the model's token-level confidence, where lower average log probabilities indicate greater uncertainty in generating correct reasoning chains, reflecting higher reasoning difficulty.

In summary, by explicitly decomposing difficulty into data-centric (**perceptual**) and model-centric (**reasoning**) groups, Durian allows each group of samples to share separate *std*s for perceptual and reasoning aspects. These normalized advantages are then combined to effectively integrate intrinsic data complexity and model uncertainty, ensuring stable optimization that preserves meaningful intra-group distinctions. To validate Durian, we conduct a comprehensive evaluation comparing it with leading methods on multiple benchmarks, and experimental results demonstrate that Durian attains more than 11.3% average performance improvements.

## 2. Preliminary

In this section, we introduce the key concepts and training setup for multimodal reasoning under RLVR (DeepSeek-AI et al., 2025). We first formulate the task, and then revisit the standard GRPO framework (Shao et al., 2024) and its improved variant, Decoupled Clip and Dynamic Sampling Policy Optimization (DAPO) (Yu et al., 2025b).

### 2.1. Task Formulation

We consider the problem of multimodal reasoning under the RLVR paradigm. Let $\{\mathcal{I}, \mathcal{Q}\} \in \mathcal{D}$ denote a multimodal input, where the dataset $\mathcal{D}$ includes image $\mathcal{I}$ and text question $\mathcal{Q}$. The model generates a reasoning response $o$ given $\{\mathcal{I}, \mathcal{Q}\}$ and receives a verifiable reward $r$ based on the correct answer $y$ (Wu et al., 2025; Wang et al., 2025a). The response $o$ typically contains both the reasoning steps and the final answer, with the reasoning steps enclosed in `<think>...</think>` and the final answer enclosed in `\boxed{}`. We employ a binary reward function, where $r(o, y) = 1$ if the final answer is equal to the correct answer $y$, and $r(o, y) = 0$ otherwise. The reasoning process is modeled as a policy $\pi_\theta(o|\mathcal{I}, \mathcal{Q})$ parameterized by $\theta$ to maximize the expected reward:

$$\mathcal{J}_{\text{RLVR}}(\theta) = \max_\theta \mathbb{E}_{\{\mathcal{I}, \mathcal{Q}\} \sim \mathcal{D}} \mathbb{E}_{o \sim \pi_\theta(\cdot|\mathcal{I}, \mathcal{Q})}[r(o, y)]. \quad (1)$$

Our goal is to enhance the reasoning capabilities of an instruction-tuned MLLM, thereby significantly improving its performance on downstream multimodal reasoning tasks.

### 2.2. Core Algorithms of Reinforcement Learning with Verifiable Reward

**Group Relative Policy Optimization (GRPO)** is derived from Proximal Policy Optimization (PPO) (Schulman et al., 2017), with the key distinction that GRPO replaces the advantage estimates obtained via Generalized Advantage Estimation (GAE) with group-relative advantages computed from a group of outputs.

Specifically, for each input $\mathcal{I}, \mathcal{Q}$, GRPO samples a group of outputs $\{o_1, o_2, \ldots, o_G\}$ from the old policy model $\pi_{\theta_{\text{old}}}$, with rollout size $G$. The advantage of the $i$-th response is computed by normalizing the rewards among the group:

$$\hat{A}_i = \frac{r_i - mean(\{r_1, r_2, \ldots, r_G\})}{std(\{r_1, r_2, \ldots, r_G\})}. \quad (2)$$

GRPO adopts a clipped objective, together with a directly imposed KL penalty term:

$$\mathcal{J}_{\text{GRPO}}(\theta) = \mathbb{E}_{(\mathcal{I}, \mathcal{Q}) \sim \mathcal{D}, \{o_i\}_{i=1}^G \sim \pi_{\theta_{\text{old}}}(o|\mathcal{I}, \mathcal{Q})} \left\{ \frac{1}{G} \sum_{i=1}^G \frac{1}{|o_i|} \sum_{t=1}^{|o_i|} \right.$$

$$\min \left[ \frac{\pi_\theta(o_{i,t} \mid \mathcal{I}, \mathcal{Q}, o_{i,<t})}{\pi_{\theta_{\text{old}}}(o_{i,t} \mid \mathcal{I}, \mathcal{Q}, o_{i,<t})} \hat{A}_{i,t}, \text{clip}\left( \frac{\pi_\theta(o_{i,t} \mid \mathcal{I}, \mathcal{Q}, o_{i,<t})}{\pi_{\theta_{\text{old}}}(o_{i,t} \mid \mathcal{I}, \mathcal{Q}, o_{i,<t})}, \right. \right. \quad (3)$$

$$\left. \left. 1 - \epsilon, 1 + \epsilon \right) \hat{A}_{i,t} \right] - \beta \mathbb{D}_{KL}(\pi_\theta \| \pi_{\text{ref}}) \right\}.$$

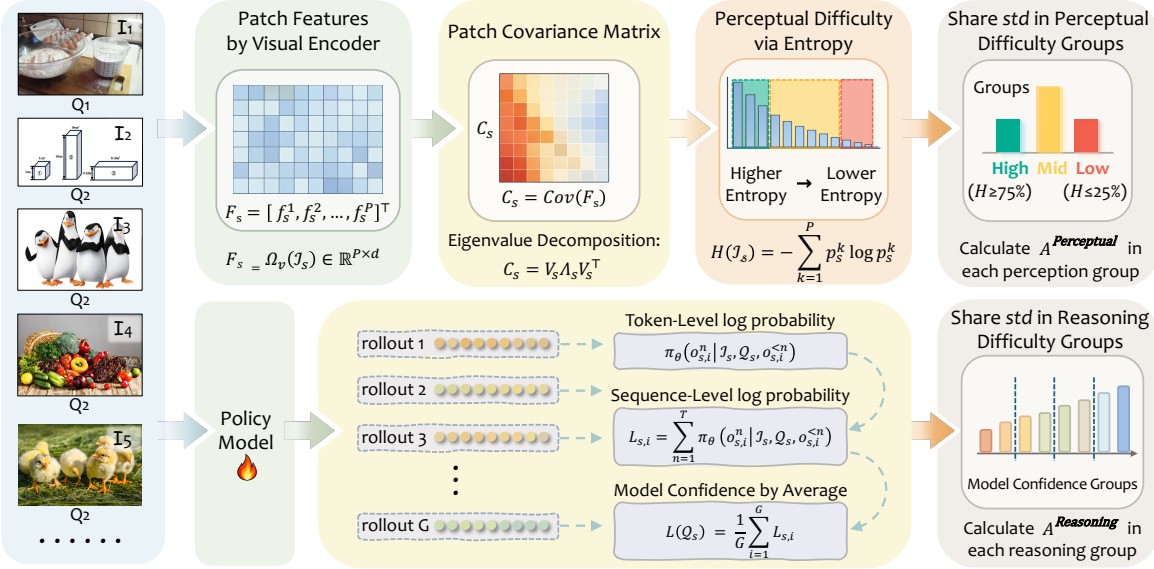

*Figure 2.* Overview of two difficulty-based regrouping strategies of Durian. Upper: For perceptual difficulty, we extract image patch features through the visual encoder and compute patch covariance matrices, whose eigenvalue entropy characterizes visual complexity. Bottom: For reasoning difficulty, model confidence is estimated from average sequence-level log probabilities across multiple rollouts. In both strategies, samples in the same group share the same *std*.

$\epsilon$ is the hyperparameter to control the clipping range of the importance sampling ratio, and $\beta$ is the penalty strength of how far the current policy $\pi_\theta$ deviates from the reference policy $\pi_{ref}$.

**Decoupled Clip and Dynamic Sampling Policy Optimization (DAPO)** is a variant of GRPO adopting an asymmetric clipping range with a larger upper bound, dynamic sampling, token-level policy gradient loss, and overlong reward shaping. The objective function of DAPO is defined as:

$$\mathcal{J}_{\text{DAPO}}(\theta) = \mathbb{E}_{(\mathcal{I},\mathcal{Q})\sim\mathcal{D},\,\{o_i\}_{i=1}^G\sim\pi_{\theta_{\text{old}}}(o|\mathcal{I},\mathcal{Q})}\left\{\frac{1}{\sum_{i=1}^G|o_i|}\sum_{i=1}^G\sum_{t=1}^{|o_i|}\right.$$

$$\min\left[\frac{\pi_\theta\left(o_{i,t}\mid\mathcal{I},\mathcal{Q},o_{i,<t}\right)}{\pi_{\theta_{\text{old}}}\left(o_{i,t}\mid\mathcal{I},\mathcal{Q},o_{i,<t}\right)}\,\hat{A}_{i,t},\,\text{clip}\left(\frac{\pi_\theta\left(o_{i,t}\mid\mathcal{I},\mathcal{Q},o_{i,<t}\right)}{\pi_{\theta_{\text{old}}}\left(o_{i,t}\mid\mathcal{I},\mathcal{Q},o_{i,<t}\right)},\right.$$

$$\left.\left.1-\epsilon_{\text{low}},1+\epsilon_{\text{high}}\right)\hat{A}_{i,t}\right]\right\}. \quad (4)$$

# 3. Durian: Difficulty-based Regrouping

In this section, we introduce our difficulty-based regrouping strategy in detail. We first represent our perceptual difficulty-based regrouping in Section 3.1, then we describe our reasoning difficulty-based regrouping in Section 3.2. The two regrouping strategies are summarized in Figure 2. Finally, we show the combination of these two strategies in Section 3.3.

## 3.1. Perceptual Difficulty-based Regrpouping

**Perceptual difficulty estimation.** To estimate the perceptual difficulty of a batch $\mathcal{B} = \{(\mathcal{I}_s, \mathcal{Q}_s)\}_{s=1}^B$, we first extract patch-level visual features from the Qwen2.5-VL-7B visual encoder $\mathbf{\Omega}_v$:

$$\mathbf{F}_s = \mathbf{\Omega}_v(\mathcal{I}_s) \in \mathbb{R}^{P\times d} = [\boldsymbol{f}_s^1, \boldsymbol{f}_s^2, \cdots, \boldsymbol{f}_s^P]^\top, \quad (5)$$

where $P$ denotes the number of spatial patches and $d$ is the feature dimension, with $\boldsymbol{f}_s^j \in \mathbb{R}^{d\times 1}, j = 1, \cdots, P$ representing the feature of the $j$-th patch.

Compared to CLIP-based representations (Radford et al., 2021), these patch-level features not only capture finer spatial granularity that preserves local details, but also align better with the downstream textual decoder $\mathbf{\Omega}_t$, ensuring both stability and semantic consistency.

We then compute the empirical covariance matrix to capture intra- and inter-patch variances:

$$\mathbf{C}_s = \tfrac{1}{P-1}\left(\mathbf{F}_s - \mathbf{1}_P\boldsymbol{\mu}_s^\top\right)\left(\mathbf{F}_s - \mathbf{1}_P\boldsymbol{\mu}_s^\top\right)^\top, \quad \boldsymbol{\mu}_s = \tfrac{1}{P}\sum_{j=1}^P\boldsymbol{f}_s^j, \quad (6)$$

where $\mathbf{1}_P \in \mathbb{R}^{P\times 1}$ is a column vector of ones, $\boldsymbol{\mu}_s \in \mathbb{R}^{d\times 1}$ is the mean patch feature of $\mathcal{I}_s$, and $\mathbf{C}_s \in \mathbb{R}^{P\times P}$ characterizes the covariance structure among spatial patches after feature centering. In particular, each diagonal entry of $\mathbf{C}_s$ measures the variance contribution of an individual patch feature relative to the mean feature, while each off-diagonal entry captures the co-variation between a pair of patches in the feature space. This patch-level covariance structure

reflects whether the visual content is concentrated in a few distinctive regions or distributed across multiple mutually correlated patches, thereby providing a principled signal for assessing perceptual difficulty.

Since $\mathbf{C}_s$ is a symmetric positive semidefinite matrix, we perform eigenvalue decomposition for spectral analysis:

$$\mathbf{C}_s = \mathbf{V}_s \mathbf{\Lambda}_s \mathbf{V}_s^\top, \quad \mathbf{\Lambda}_s = \mathrm{diag}(\lambda_s^1, \ldots, \lambda_s^P), \ \lambda_s^k \geq 0. \quad (7)$$

$\lambda_s^k$ denotes the $k$-th eigenvalue, quantifying the variance along one orthogonal principal direction. Concentrated eigenvalues indicate that most variance is captured by a few dimensions, whereas more balanced eigenvalues imply richer visual structure and higher visual complexity.

The final perceptual difficulty score is defined as the entropy (Shannon, 1948) of the normalized distribution of eigenvalues:

$$H(\mathcal{I}_s) = -\sum_{k=1}^{P} p_s^k \log p_s^k, \quad (8)$$

where $p_s$ is the normalized probability distribution of the eigenvalues, and each element in $p_s$ is calculated as:

$$p_s^k = \frac{\lambda_s^k}{\sum_{j=1}^{P} \lambda_s^j}, \quad \text{with} \sum_{k=1}^{P} p_s^k = 1. \quad (9)$$

Here, low entropy corresponds to the visually easy sample, with variance concentrated on a few dominant components, whereas high entropy indicates the visually difficult sample, with variance distributed across many patches.

**Perceptual difficulty-based regrouping.** Given the perceptual difficulty scores within a batch, we partition samples into three groups using the 25th and 75th percentiles $\tau_{0.25}$ and $\tau_{0.75}$:

$$\begin{cases} \mathcal{S}_1 = \{s \mid H(\mathcal{I}_s) \leq \tau_{0.25}\}, \\ \mathcal{S}_2 = \{s \mid \tau_{0.25} < H(\mathcal{I}_s) < \tau_{0.75}\}, \\ \mathcal{S}_3 = \{s \mid H(\mathcal{I}_s) \geq \tau_{0.75}\}. \end{cases} \quad (10)$$

For each group $a$, the reward set can be defined as:

$$\mathcal{R}_a = \{r_{s,i} \mid i = 1, \cdots, G, s \in \mathcal{S}_a\}, \ a \in \{1, 2, 3\}, \quad (11)$$

where $r_{s,i}$ refers to the $i$-th reward of the $s$-th sample which belongs to the group $\mathcal{S}_a$.

We can then compute the shared standard deviation $std(\mathcal{R}_a)$ of group rewards, and normalize the reward of each sample in batch with the new $std(\mathcal{R}_a)$ to calculate advantage accordingly:

$$A_{s,i}^{\text{Perceptual}} = \frac{r_{s,i} - mean(r_{s,1}, r_{s,2}, \ldots r_{s,G})}{std(\mathcal{R}_a)}, \quad (12)$$

$$std(\mathcal{R}_a) = \sqrt{\frac{1}{|\mathcal{R}_a| - 1} \sum_{r_{s,i} \in \mathcal{R}_a} \left(r_{s,i} - \frac{1}{|\mathcal{R}_a|} \sum_{r_{s,i} \in \mathcal{R}_a} r_{s,i}\right)^2}, \quad (13)$$

where $|\mathcal{R}_a|$ denotes to the cardinality of $\mathcal{R}_a$.

By grouping samples into low-, medium-, and high-entropy categories, the normalization scale is shared only among samples with comparable perceptual difficulty. This mitigates the influence of extreme samples, balances treatment across different levels of visual complexity, and ultimately stabilizes optimization.

### 3.2. Reasoning Difficulty-based Regrouping

**Reasoning difficulty estimation.** While perceptual difficulty captures the intrinsic complexity of the image, reasoning difficulty is shaped by the model's intrinsic confidence in generating the final answer. Even for inputs with similar visual complexity, the model may exhibit varying confidence levels: high confidence (assigning a high probability to reasoning chains) implies a clear and reliable reasoning path, whereas low confidence indicates uncertainty and potential reasoning failures. Following this intuition, we quantify reasoning difficulty using the model's probabilities for its reasoning chains.

For the given batch $\mathcal{B} = \{(\mathcal{I}_s, \mathcal{Q}_s)\}_{s=1}^{B}$, and the generated $G$ responses for each sample, we denote the $i$-th response as $o_{s,i} = (o_{s,i}^1, \ldots, o_{s,i}^T)$, where $o_{s,i}^n$ is the $n$-th token and $T$ is the sequence length.

Based on token-level log probability $\pi_\theta\left(o_{s,i}^n \mid \mathcal{I}_s, \mathcal{Q}_s, o_{s,i}^{<n}\right)$, we aggregate across tokens to obtain the sequence-level log probability for response $o_{s,i}$:

$$L_{s,i} = \sum_{n=1}^{T} \log \pi_\theta\left(o_{s,i}^n \mid \mathcal{I}_s, \mathcal{Q}_s, o_{s,i}^{<n}\right). \quad (14)$$

Then we define model confidence for sample $(\mathcal{I}_s, \mathcal{Q}_s)$ as the average sequence-level log probability across its $G$ rollouts:

$$L(\mathcal{Q}_s) = \frac{1}{G} \sum_{i=1}^{G} L_{s,i}. \quad (15)$$

This formulation reflects the model's internal confidence: High and consistent $L(\mathcal{Q}_s)$ indicates reliable reasoning chains, whereas low or fluctuating $L(\mathcal{Q}_s)$ reflects epistemic uncertainty, implying a more challenging reasoning sample.

**Reasoning difficulty-based regrouping.** Given the model confidence scores $L(\mathcal{Q}_s)$ for the batch $\mathcal{B} = \{(\mathcal{I}_s, \mathcal{Q}_s)\}_{s=1}^{B}$, we divide samples into $b$ groups according to the quantiles of their confidence distribution. Let $\{\tau_0, \tau_1, \ldots, \tau_b\}$ denote the quantile boundaries, with $\tau_0 = 0$ and $\tau_b = 1$. Each question $\mathcal{Q}_s$ is then assigned to a group by:

$$\mathcal{M}_u = \{s \mid \tau_{u-1} \leq L(\mathcal{Q}_s) < \tau_u\}, \quad u \in \{1, \ldots, b\}. \quad (16)$$

Within each group $\mathcal{M}_u$, we define the reward set as:

$$\mathcal{R}_u = \{r_{s,i} \mid i = 1, \dots, G, s \in \mathcal{M}_u\}, \ u \in \{1, \dots, b\}, \quad (17)$$

where $r_{u,i}$ is the reward of the $i$-th response for the sample, which belongs to the $u$ group. We can then calculate the shared standard deviation $std(\mathcal{R}_\mathcal{W})$ of reasoning difficulty-based group, and compute the advantage accordingly:

$$A_{s,i}^{\text{Reasoning}} = \frac{r_{s,i} - mean(r_{s,1}, r_{s,2}, \dots r_{s,G})}{std(\mathcal{R}_u)}, \quad (18)$$

where the *std* can be calculated as:

$$std(\mathcal{R}_u) = \sqrt{\frac{1}{|\mathcal{R}_u| - 1} \sum_{r_{s,i} \in \mathcal{R}_u} \left(r_{s,i} - \frac{1}{|\mathcal{R}_u|} \sum_{r_{s,i} \in \mathcal{R}_u} r_{s,i}\right)^2}, \quad (19)$$

This regrouping ensures that responses with similar confidence levels are normalized on comparable scales, mitigating instability introduced by overconfident or underconfident samples.

### 3.3. Combination for Robust Optimization

To leverage the complementary aspects of perceptual and reasoning difficulty, we propose an element-wise combination strategy. Specifically, given the perceptual-based group normalized advantage $A^{\text{Perceptual}}$, the reasoning-based group normalized advantage $A^{\text{Reasoning}}$, and the original GRPO advantage $A^{\text{GRPO}}$, the combined advantage is defined as:

$$
\begin{aligned}
A^{\text{Combined}} = & \ \alpha_{\text{Ori}} \cdot A^{\text{GRPO}} \\
& + \alpha_{\text{Percep}} \cdot A^{\text{Perceptual}} \\
& + \alpha_{\text{Reason}} \cdot A^{\text{Reasoning}},
\end{aligned} \quad (20)
$$

where $\alpha_{\text{Ori}}, \alpha_{\text{Percep}}, \alpha_{\text{Reason}}$ are weighting coefficients that balance the contributions of the three components. Perceptual difficulty, quantified by the entropy in the image, captures the *visual complexity* of multimodal inputs; reasoning difficulty, derived from token- and sequence-level log probabilities, reflects the *model uncertainty* during reasoning. Integrating these difficulty-based advantages with the original GRPO advantage allows the model to preserve meaningful intra-sample distinctions and incorporate both intrinsic and extrinsic difficulty context, providing a more stable and informative advantage for policy optimization.

## 4. Experiment

In this section, we conduct comprehensive experiments to address the following research questions:

- RQ1: How does Durian perform on multimodal reasoning tasks compared to other baseline methods?
- RQ2: How do key components of Durian influence its performance?

- RQ3: How is the sensitivity of Durian under varying hyperparameters?
- RQ4: What is the computational overhead of Durian compared with vanilla GRPO?

### 4.1. Experimental Settings

**Dataset.** For training, we rely on the Geometry3K (Lu et al., 2021) dataset, which provides 2.1K training samples and 0.3K validation samples. Besides, we also provide experimental results training on a larger dataset ViRL39k.

**Benchmark.** We evaluate Durian on five benchmarks: four visual reasoning datasets, namely MathVerse (Zhang et al., 2024), MathVision (Wang et al., 2024), MathVista (Lu et al., 2024b), and WeMath (Qiao et al., 2025), as well as one visual perception benchmark, HallusionBench (Guan et al., 2024). In addition, we assess the in-domain performance by comparing Durian with the vanilla GRPO and DAPO.

**Baseline.** To evaluate the performance of Durian, we consider three categories of baselines: (1) Closed-source models: GPT-4o (Aaron et al., 2024), and Cloud-3.5-sonnet (Anthropic, 2024). (2) Open source models: InternVL-2.5-8B-Instruct (Chen et al., 2024), LLaVA-OneVision-7B (Li et al., 2024), Kimi-VL-16B (Du et al., 2025a), URSA-8B (Luo et al., 2025), and Mulberry-7B (Yao et al., 2024). (3) RLVR-based Models: MLLMs trained with reinforcement learning using verifiable rewards, representing the current mainstream approaches in this line of research. This category includes R1-VL-7B (Zhang et al., 2025a) , Vision-R1-7B (Huang et al., 2025b) , R1-OneVision-7B (Yang et al., 2025b) , OpenVLThinker-7B (Deng et al., 2025) , MM-Eureka-Qwen-7B (Meng et al., 2025) , ADORA-7B (Gui & Ren, 2025) , ThinkLite-7B-VL (Wang et al., 2025d), and VLAA-Thinker-7B (Chen et al., 2025a).

**Implementation details.** Following prior work (Liu et al., 2025a), we use Qwen2.5-VL-7B (Bai et al., 2025) as base model and adpot EasyR1 (Zheng et al., 2025) as reinforcement learning framework. All experiments are conducted on 8 NVIDIA H20 96G GPUs. We adopt the default settings from EasyR1, using a learning rate of $1e^{-6}$, a global batch size of 128, a rollout batch size of 512, and a rollout size of 8. The analysis of rollout size is provided in Appendix E.

**Evaluation.** We follow the same evaluation protocol as NoisyRollout (Liu et al., 2025a). Specifically, we use the evaluation suite developed by NoisyRollout for consistent assessment of our trained checkpoints and most open-source R1-related checkpoints using vLLM for accelerated inference. We also employ greedy decoding for model inference and use Gemini-2.0-Flash-001 (Team, 2023) as the judge model to parse generated responses. For the methods and models both compared by NoisyRollout and us,

*Table 1.* Performance comparison of Multi-modal LLMs with over 5 benchmarks. Accuracy scores (%) are reported for all benchmarks for clarity. Data sizes used for SFT and RL are annotated in blue and red, respectively. The best value in each column is shown in **bold**, and the second-best is underlined.

| Model | Data Size | MathVerse | MathVision | MathVista | WeMath | HallusionBench | Average |
|---|---|---|---|---|---|---|---|
| *Close-source models* | | | | | | | |
| GPT-4o | - | 50.8 | 30.4 | 63.8 | 69.0 | 71.4 | - |
| Claude-3.5-Sonnet | - | 26.5 | 38.0 | 67.7 | - | 71.6 | - |
| *Open-source models* | | | | | | | |
| InternVL-2.5-8B-Instruct (Chen et al., 2024) | - | 39.5 | 19.7 | 64.4 | - | 67.3 | - |
| LLaVA-OneVision-7B (Li et al., 2024) | - | 26.2 | - | 63.2 | - | 48.4 | - |
| Kimi-VL-16B (Du et al., 2025a) | - | 44.9 | 21.4 | 68.7 | - | 66.2 | - |
| URSA-8B (Luo et al., 2025) | - | 45.7 | 26.2 | 59.8 | - | - | - |
| Mulberry-7B (Yao et al., 2024) | - | - | - | 63.1 | - | - | - |
| *reinforcement learning with verifiable reward based* | | | | | | | |
| R1-VL-7B (Zhang et al., 2025a) | 260K+10K | 52.2 | 28.2 | 74.3 | 69.0 | 57.2 | 56.2 |
| Vision-R1-7B (Huang et al., 2025b) | 200K+10K | 52.4 | 27.2 | **73.5** | 62.9 | 69.2 | 57.0 |
| R1-OneVision-7B (Yang et al., 2025b) | 155K+10K | 46.1 | 22.5 | 63.9 | 62.1 | 65.6 | 52.0 |
| OpenVLThinker-7B (Deng et al., 2025) | 35K+15K | 48.0 | 25.0 | 71.5 | 67.8 | 70.8 | 56.6 |
| MM-Eureka-Qwen-7B (Meng et al., 2025) | 15K | 50.5 | 28.3 | 71.5 | 65.5 | 68.3 | 56.8 |
| ADORA-7B (Gui & Ren, 2025) | 2.1K | 50.1 | 27.6 | 71.1 | 67.1 | 53.1 | 53.8 |
| ThinkLite-7B-VL (Wang et al., 2025d) | 11K | 50.2 | 27.6 | 72.7 | 69.2 | 71.0 | 58.1 |
| VLAA-Thinker-7B (Chen et al., 2025a) | 25K | 49.9 | 26.9 | 68.8 | 67.9 | 68.6 | 56.4 |
| NoisyRollout (Liu et al., 2025a) | 2.1K | **53.2** | 28.5 | 72.6 | 69.6 | 72.1 | 59.2 |
| Qwen2.5-VL-7B-Instruct (Bai et al., 2025) | - | 46.2 | 25.0 | 67.5 | 63.1 | 64.6 | 53.3 |
| + Vanilla GRPO | 2.1K (Geometry3K) | 49.6 | 26.8 | 70.2 | 68.2 | 69.8 | 56.9 |
| + Durian (based on Vanilla GRPO) | 2.1K (Geometry3K) | 52.8 | 28.8 | 72.3 | 69.2 | **72.9** | 59.2 |
| + Vanilla DAPO | 2.1K (Geometry3K) | 50.4 | 27.6 | 70.7 | 69.4 | 68.6 | 57.3 |
| + Durian (based on Vanilla DAPO) | 2.1K (Geometry3K) | 51.9 | **29.0** | 72.2 | **71.8** | 71.4 | **59.3** |
| + Durian (based on Vanilla DAPO) | 39K (ViRL39K) | 52.4 | 29.9 | 73.8 | 72.0 | 72.5 | 60.1 |

we directly report the results from NoisyRollout instead of re-evaluating.

## 4.2. Comparison with Baseline Methods (RQ1)

We comprehensively compare Durian with various state-of-the-art methods, and experimental results are listed in Table 1. We can draw the following observations: (1) Compared with those either distilled from large-scale chain-of-thought data or employing complex data augmentation strategies, our method, utilizing only 2.1k training samples, achieves comparable or even superior performance, significantly demonstrating our effectiveness. (2) Building upon both GRPO and DAPO, our strategy demonstrates promising performance gains. Specifically, we achieve an average of 11.3% improvements over Qwen2.5-VL, especially on Mathvision, our strategy achieves more than 16% improvements, further showing our effectiveness.

## 4.3. Ablation Studies (RQ2)

To better understand the contribution of each component in Durian, we conduct ablation studies on five benchmarks, comparing four settings over Qwen2.5-vl: vanilla DAPO, DAPO with perceptual regrouping, DAPO with reasoning regrouping, and our Durian. Results are in Figure 3.

**The effects of Perceptual difficulty-based regrouping.**
Using perceptual difficulty-based regrouping alone yields consistent performance gains across benchmarks. For instance, on HallusionBench, which is explicitly designed to evaluate perceptual ability, we observe an improvement of 3.4% over vanilla DAPO. This demonstrates that regrouping samples via spectral analysis of image patch covariances enhances the model's perceptual grounding by mitigating the dominance of extremely easy or hard cases.

**The effects of Reasoning difficulty-based regrouping.**
The average accuracy under model confidence regrouping increases to 58.4, and it's notable to observe a 3.8% gain on MathVerse, even surpassing the performance of our method, indicating that the model's internal confidence estimation also serves as a reliable signal for stabilizing optimization.

The combination of both strategies achieves the best overall performance with an average accuracy of 59.3. This confirms that these two strategies provide complementary perspectives on samples, and their integration leads to more robust policy optimization.

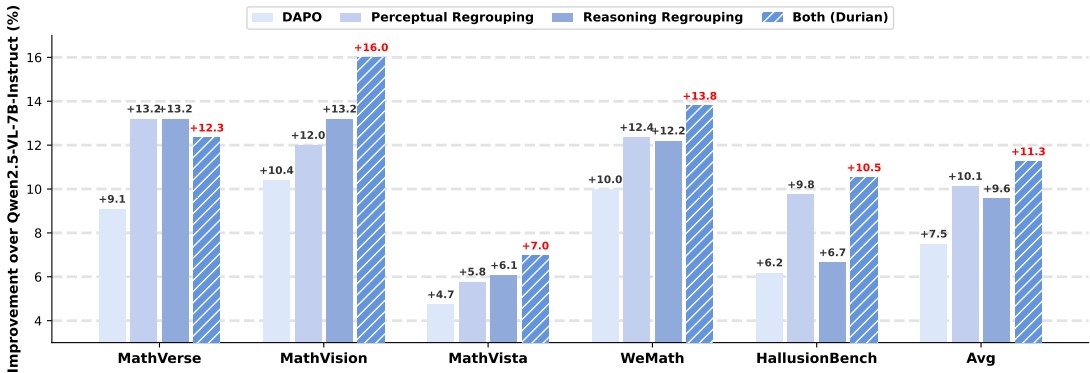

*Figure 3.* Acc Improvements of two re-grouping strategies over Qwen2.5-VL. We take DAPO as our backbone.

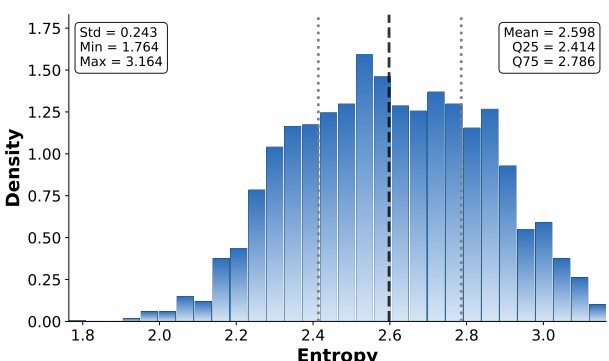

*Figure 4.* The distribution of pre-calculated entropy on the Geometry3K. $x$ axis represents entropy, $y$ axis is the probability density, Q25 and Q75 denotes 25th and 75th percentiles, respectively .

### 4.4. Hyper-parameter Sensitivity Analysis (RQ3)

In this section, we analyze the effects of hyperparameters, including the number of perceptual, reasoning difficulty-based groups, and $\alpha_{\text{Ori}}, \alpha_{\text{Percep}}, \alpha_{\text{Reason}}$.

#### 4.4.1. GROUPS UNDER PERCEPTUAL DIFFICULTY-BASED STRATEGY

As shown in Figure 4, to regroup samples by entropy, we adopt the 25th and 75th percentiles as thresholds. This quantile-based choice is inherently distribution-aware, as it adapts to the empirical spread of entropy values rather than relying on arbitrary fixed cutoffs. It produces a natural 1:2:1 partition of the data—approximately 25% easy, 50% medium, and 25% hard—avoiding the issue of overly sparse or dense categories. Such a balance is desirable for stable optimization: each group contains sufficient samples to provide reliable intra-group statistical estimates, while extremely low- and high-entropy cases are isolated rather than allowed to dominate normalization. Moreover, this three-level categorization is semantically interpretable, with low entropy corresponding to simple scenes, high entropy to complex ones, and the middle range capturing moderately

*Table 2.* The accuracy performance of different numbers of groups $b$ on five benchmarks.

| Groups $b$ | MathVerse | MathVision | MathVista | WeMath | HallusionBench |
|---|---|---|---|---|---|
| 4 | 49.4 | 28.1 | 68.6 | 66.7 | 66.8 |
| 6 | 50.6 | 27.1 | 69.7 | 70.2 | 70.0 |
| 10 | 50.6 | 27.5 | 71.3 | 70.8 | 68.9 |
| 12 | **52.3** | **28.5** | 70.9 | **71.4** | 69.6 |
| 16 | 50.4 | 27.5 | **72.4** | 70.7 | 69.9 |
| 20 | 51.5 | 28.1 | 71.3 | 70.4 | 68.8 |
| 24 | 50.1 | 27.9 | 71.6 | 70.1 | 69.7 |
| 32 | 50.5 | 26.9 | 71.6 | 68.7 | 70.7 |
| 40 | 50.7 | 28.3 | 71.4 | 70.2 | 70.3 |
| 48 | 50.0 | 27.7 | 71.5 | 69.5 | **70.9** |

difficult cases (For further empirical analysis, see Appendix B). Detailed cases representing the entropy of these three categories are illustrated in Appendix G.1.

#### 4.4.2. GROUPS UNDER REASONING DIFFICULTY-BASED STRATEGY

We investigate the impact of varying the number of groups in the reasoning-based regrouping strategy in Table 2. We observe that the performance is relatively stable across a wide range of groups, suggesting that our method is robust to this hyperparameter. For instance, on MathVista and HallusionBench, the accuracy is steadily improved as the number of groups increases to around 12–16, after which the results plateau with only minor fluctuations. This suggests that moderate group granularity is sufficient to capture meaningful variations in reasoning difficulty, while overly fine partitioning provides diminishing returns. A similar trend is observed on WeMath, where the performance peaks at 12 groups but remains competitive without significant degradation even when more groups are introduced.

#### 4.4.3. ANALYSIS OF DIFFERENT WEIGHTING COEFFICIENTS.

We experiment with different combinations of three coefficients $\alpha_{Ori}$, $\alpha_{Percep}$, and $\alpha_{Reason}$ in Table 3. We can

observe that while the performance on different benchmarks varies slightly with different settings, the method is relatively stable across a wide range of settings, with no significant degradation in results, indicating that our model is not overly sensitive to the specific choice of hyperparameters. This suggests that our method does not require extremely fine-tuned hyperparameters to perform effectively.

*Table 3.* The effects of different weighting coefficients (built upon DAPO) on 5 benchmarks

| $\alpha_{\text{Ori}}$ | $\alpha_{\text{Percep}}$ | $\alpha_{\text{Reason}}$ | MathVerse | MathVision | MathVista | WeMath | HallusionBench |
|------|------|------|------|------|------|------|------|
| 0.1 | 0.2 | 0.7 | 50.7 | 28.6 | 71.6 | 71.8 | 71.2 |
| 0.15 | 0.25 | 0.6 | 50.8 | 29.0 | 71.6 | 70.6 | 70.8 |
| 0.2 | 0.1 | 0.7 | 51.2 | 28.4 | 71.5 | 70.4 | 71.0 |
| 0.3 | 0.1 | 0.6 | 51.7 | 27.8 | 70.7 | 70.6 | 70.8 |
| 0.4 | 0.3 | 0.3 | 51.9 | 27.9 | 71.4 | 70.5 | 71.1 |
| 0.6 | 0.2 | 0.2 | 50.4 | 28.8 | 72.2 | 71.0 | 71.4 |
| 0.7 | 0.1 | 0.2 | 51.1 | 28.3 | 70.3 | 71.1 | 69.8 |

### 4.5. Computational Cost

A natural concern is whether the difficulty-aware regrouping in Durian introduces additional training overhead. We analyze this question from two perspectives: (i) what extra computation is actually added, (ii) end-to-end wall-clock time compared with vanilla GRPO.

**What is added.** Durian introduces no additional trainable module, and no extra policy-model forward pass. The added cost comes only from two lightweight statistics for regrouping: (i) Perceptual difficulty. The eigenvalue entropy in Eq. 8 depends solely on the image and the frozen visual encoder $\Omega_v$, and is independent of the policy parameters $\theta$. We therefore pre-compute it once offline before training and cache the resulting score in the dataset, so it is never recomputed across training steps. (ii) Reasoning difficulty. The sequence-level log-probabilities $L_{s,i}$ in Eq. 14 are already produced as a by-product of rollout generation. Durian only adds a 'torch.quantile'-based binning step over these values to assign each sample to one of $b$ groups. This step involves no forward pass and no parameter update.

**End-to-end wall-clock cost.** The offline perceptual-difficulty pre-computation finishes in about 15 minutes, which is negligible compared with roughly 9 hours for one full GRPO/DAPO training run on 8 NVIDIA H20 GPUs. Within each training step, the binning step contributes a sub-second overhead. As a result, the total training time of Durian (DAPO) is on par with vanilla DAPO, while delivering the gains reported in Table 1.

## 5. Related Works

In this section, we overview of the related studies (Fang et al., 2024a; Lu et al., 2025a;b; 2023). Specifically, we first discuss representative strategies to construct multimodal reasoning models, including chain-of-thought distillation,

reinforcement learning, and visual tool integration. Then we introduce the RLVR and its optimization variants, and finally, we highlight the key differences between ours and existing approaches.

### 5.1. Multimodal Reasoning Models

**Chain-of-thought distillation.** Supervised fine-tuning on long CoT data enables models to learn detailed reasoning traces, thereby improving reasoning accuracy. Specifically, building upon (Zhang et al., 2023), this strategy has proven effective through both transferring CoT-enhanced LLMs to multimodal settings (Du et al., 2025b) and training directly with multimodal reasoning data (Liu et al., 2023). Recent works explore different forms of intermediate reasoning supervision (Dai et al., 2023; Yang et al., 2025c; Fang et al., 2026; 2023; 2024b; Li et al., 2025a; Fang et al., 2024c).

**Reinforcement learning.** Another line of research leverages RL to optimize reasoning trajectories beyond imitation. Most studies adopt PPO (Schulman et al., 2017) or GRPO (Shao et al., 2024), with representative approaches such as (Shen et al., 2025; Wang et al., 2025b;c) that apply RL across diverse domains. We will elaborate on RLVR and GRPO in the following subsection (Section 5.2).

**Visual tool integration.** This paradigm moves beyond merely "thinking about images" toward actively querying, modifying, and generating visual information as intermediate steps in reasoning, forming a "visual chain of thought". The development of think-with-image can be roughly divided into three stages (Su et al., 2025): from external tool exploration (Ma et al., 2024; 2025), through programmatic manipulation (Surís et al., 2023; Fu et al., 2025), to intrinsic imagination (Zhao et al., 2025; Chen et al., 2025b). These three stages reflect interconnected capabilities—active exploration, structured reasoning, and generative planning—that together transform visual representations from static inputs into a dynamic workspace for thought.

### 5.2. Reinforcement Learning with Verifiable Reward

RLVR (Lambert et al., 2024) is an optimization paradigm that replaces subjective reward scores with verifiable signals. Its core algorithm, GRPO (Shao et al., 2024), stabilizes training by comparing candidate responses within a group. Subsequent studies can be broadly categorized into two directions: data-centric methods, which expand the candidate and reward space through data manipulation or augmentation, and algorithm-centric methods, which refine GRPO to strengthen semantic grounding and coherent reasoning.

**Data-centric GRPO.** Recent advances (Fang et al., 2025; Lu et al., 2026a;b; 2024a), especially this line of work, enlarge the candidate set (Chen et al., 2025d) or restructure the training data (Chen et al., 2025c; Zhu et al., 2025) so

that group comparisons capture richer behaviors. By manipulating data distributions (Zhu et al., 2025) or augmenting inputs (Li et al., 2025b; Liu et al., 2025a), these methods expose models to a wider variety of responses, thereby increasing the likelihood of discovering high-quality verifiable signals.

**Algorithm-centric GRPO.** In contrast, algorithm-centric methods refine how verifiable signals guide reasoning. Rather than expanding candidate sets, they adapt GRPO to enhance semantic grounding (Yu et al., 2025a; Liu et al., 2025c) and logical coherence (Huang et al., 2025a; Wei et al., 2025). These approaches emphasize the role of visual grounding and promote reasoning chains where intermediate steps remain verifiable while supporting the final answer.

**Difference.** Compared with existing methods, we regroup the data in advantage calculation based on model response uncertainty and the entropy of images when computing *std*, and share the *std* within each group. This design prevents the model from overfitting to extreme samples and enhances its ability to capture the data distinction within each group.

## 6. Conclusion

In this work, we identify a critical challenge in GRPO-based reinforcement learning methods for multimodal reasoning tasks: **the *std*-based group normalization is sensitive to extreme samples**, such as response groups that are almost entirely positive or negative. While this issue exists in GRPO in general, it is significantly amplified in multimodal settings due to the joint influence of perceptual complexity and reasoning uncertainty.

To address this, we propose Durian, an effective **difficulty-aware re-grouping** strategy. By decomposing the difficulty into perceptual and reasoning aspects, we construct groups of samples with similar difficulty levels, allowing each group to share *std* during normalization. The normalized advantages with shared *std* from both aspects are combined via element-wise combination, effectively integrating data complexity and model uncertainty while preserving intra-group distinctions. By applying over GRPO and DAPO, our strategy achieves 11.3% average performance gains across multiple multimodal reasoning benchmarks.

**Limitations and Future Work.** Durian inevitably introduces some hyperparameters, but we observe that performance across benchmarks remains relatively stable under a wide range of settings. This empirical robustness suggests that Durian is not overly sensitive to precise hyperparameter tuning. Several directions remain open for future work, such as more precise difficulty estimation and adaptive grouping strategies, which may better balance intra-group distinctions and mitigate extreme samples. Beyond technical refinements, the underlying principle of aligning optimization with sample difficulty also offers a general paradigm for stabilizing RL optimization with multimodal inputs.

## Acknowledgements

This research is supported by the National Natural Science Foundation of China under Grants No. 62572449 and U24B20180.

## Impact Statement

In our view, the main impact of this work is not only the performance gain itself, but also the perspective it offers to the community: as multimodal reasoning models are trained beyond RLVR objectives, the design of normalization and optimization mechanisms becomes as important as model scale or data scale. Our work shows that sample difficulty and normalization design can materially affect whether RL training amplifies useful learning signals or overfits to extreme reward patterns. We hope this perspective encourages future work on more robust, difficulty-aware, and theoretically grounded post-training methods for MLLMs.

At the same time, we emphasize that stronger benchmark performance does not by itself guarantee safety, fairness, or robustness in real-world deployment. Improved reasoning ability may also make incorrect or misleading outputs appear more convincing. Therefore, we believe such models should still be evaluated with domain-specific safeguards, robust error analysis, and appropriate safety controls before being used in high-stakes settings.

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

## A. Analysis of the occurrence of extreme samples, which *std*-based group normalization is highly sensitive to.

To further support the motivation of our paper and analyze the changes of extreme samples during training, we perform a detailed step-by-step analysis of reward statistics across 60 training steps with 512 samples and their 8 rollout rewards. The empirical evidence clearly shows that the existence of extreme samples is not an occasional event but a persistent and systemic phenomenon.

*Table 4.* The statistics of rewards about extreme samples within the batch across 60 training steps.

| Training steps | 1 | 10 | 20 | 30 | 40 | 50 | 60 |
|---|---|---|---|---|---|---|---|
| Effective samples (participating in training) | 323 | 327 | 324 | 322 | 297 | 314 | 306 |
| Extreme success (7 correct & 1 wrong) | 41 | 39 | 48 | 66 | 78 | 60 | 82 |
| Extreme failure (7 wrong & 1 correct) | 78 | 89 | 74 | 51 | 54 | 54 | 51 |
| Total Extreme Ratio | 36.8% | 39.1% | 37.7% | 36.3% | 44.4% | 36.31% | 43.5% |

First, groups with 8 identical rewards (i.e., variance = 0) constitute 35%–46% of all samples at every training step. We first exclude these groups for not participating in gradient updates.

Second, during the training process, there are **31%–44% samples that exhibit the 7:1 extreme reward patterns** (i.e., 7/8 correct or wrong) among the remaining effective samples, which produce extremely small variance. Besides, the occurrence of this situation will increase as training deepens.

These findings demonstrate that **the instability of *std*-based normalization is structural rather than incidental:** multimodal reasoning tasks naturally contain a large proportion of very easy and very hard samples, leading to unstable and unreliable advantage scaling. This directly motivates our difficulty-aware regrouping strategy, which stabilizes normalization by ensuring that variance is computed only within samples of comparable difficulty.

## B. Verify the feasibility of utilizing image entropy as a proxy for perceptual difficulty and model confidence as a proxy for reasoning difficulty.

### B.1. Image entropy as a proxy for perceptual difficulty.

**Perceptual difficulty** in our framework is defined based on **the complexity of visual embeddings**, which we quantify using spectral analysis of image patch covariances. Specifically, the entropy of the eigenvalue distribution from the covariance matrix reflects the amount of variance across spatial features in the image. Researchers in prior works(Grzywacz, 2025) support that: high entropy indicates a more diverse distribution of visual features, implying a richer and more complex visual structure. This complexity makes it more challenging for the visual model to recognize, and thus we associate higher entropy with greater perceptual difficulty.

### B.2. Model confidence as a proxy for reasoning difficulty.

Researchers in (Farquhar et al., 2024; Nguyen et al., 2025) propose that "one measure of uncertainty is the predictive entropy of the output distribution, which measures the information one has about the output given the input[3]. The predictive entropy for an input sentence $\mathbf{x}$ is the conditional entropy ($H$) of the output random variable ($Y$) with realization $y$ given $\mathbf{x}$."

$$\text{PE}(\mathbf{x}) = H(Y|\mathbf{x}) = -\sum_{y} P(y|\mathbf{x})\ln P(y|\mathbf{x}). \tag{21}$$

Researchers(Kadavath et al., 2022) also hypothesize that when a model knows the answer to a particular question, it is confident in its response, and this would result in an answer distribution with small entropy. Conversely, when a model is unsure about its response, it will lead to an answer distribution with high entropy, thus implying a more challenging reasoning process.

This aligns directly with our formulation: the **sequence-level log probabilities** we compute are theoretically linked to the notion of *semantic entropy* and represent the joint likelihood of the entire reasoning chain. A low log-probability corresponds to a flat or high-entropy output distribution, reflecting uncertainty in the reasoning trajectory, while a high log-probability corresponds to a confident, low-entropy distribution.

### B.3. Empirical validation.

During the evaluation stage, we conduct an analysis focusing on the **questions that the model answered incorrectly** on two benchmarks. We want to examine whether these error samples are concentrated in the more difficult groups as defined by our difficulty metrics. The intuition behind this approach is that samples belonging to higher-difficulty groups—whether in terms of perceptual complexity or reasoning uncertainty—should naturally be harder for the model to tackle. Consequently, we expect these samples to exhibit higher error rates.

To achieve this, we use Gemini2.5 Pro to classify the sources of errors, distinguishing between **perceptual errors** and **reasoning errors**.

• For **perceptual errors**, we first group the images based on their visual entropy, then compute the proportion of incorrect answers within each group relative to the total number of perceptual errors.

• Similarly, for **reasoning errors**, we group the samples based on model confidence, and calculate the proportion of incorrect answers in each group relative to the total number of reasoning errors.

*Table 5.* The error rate of perceptual difficulty groups in perceptual errors on two benchmarks.

|  | low-entropy | medium-entropy | high-entropy |
|---|---|---|---|
| Wemath | 23.6% | 31.3% | 45.1% |
| HallusionBench | 21.2% | 29.6% | 49.2% |

*Table 6.* The error rate of reasoning difficulty groups in reasoning errors on two benchmarks.

|  | group 1 (low confidence) | group 2 | group 3 | group 4 | group 5 | group 6 | group 7 | group 8 | group 9 | group 10 (high confidence) |
|---|---|---|---|---|---|---|---|---|---|---|
| Wemath | 13.4% | 12.6% | 13.2% | 12.7% | 11.6% | 9.1% | 9.7% | 7.2% | 6.7% | 4.2% |
| HallusionBench | 12.7% | 11.9% | 11.2% | 11.7% | 10.0% | 9.8% | 9.0% | 8.2% | 8.3% | 7.1% |

As shown in Table 5 and Table 6, the results align with our expectations: images with **low visual entropy** (indicating simplicity) correspond to **lower perceptual error rates**, and samples with **lower model confidence** (indicating greater uncertainty in the reasoning process) correspond to **higher reasoning error rates**. Our empirical findings are consistent with this intuition, further supporting the validity of our difficulty metrics.

## C. Experiment Settings

**Reward Calculation.** We adopt a combination of format reward and accuracy reward as the final reinforcement learning signal. The two components are defined as follows:

$$r_{\text{format}} = \begin{cases} 1, & \text{if the output format is correct,} \\ 0, & \text{otherwise,} \end{cases} \tag{22}$$

$$r_{\text{acc}} = \begin{cases} 1, & \text{if the answer matches the ground truth,} \\ 0, & \text{otherwise.} \end{cases} \tag{23}$$

The overall reward is computed as the weighted sum of the two:

$$r_{\text{overall}} = 0.1 \times r_{\text{format}} + 0.9 \times r_{\text{acc}}. \tag{24}$$

A smaller weight is assigned to the format reward, since response formatting is relatively easy to learn compared with accuracy.

## D. Prompt Design

We use a "Thinking prompt" to formalize the output of the model. It requires the model to put its reasoning process within `<think>...</think>` and the final answer in `\boxed{}`. We keep the system prompt of Qwen2.5-VL (Bai et al., 2025) and prepend the "Thinking prompt" to the user message. The same format is used for both training and evaluation. The full instruction prompt is as follows:

---

**Prompt Example**

**SYSTEM:**
You are a helpful assistant.
**USER:**
You FIRST think about the reasoning process as an internal monologue and then provide the final answer. The reasoning process MUST BE enclosed within `<think>...</think>` tags. The final answer MUST BE put in `\boxed{}`.<QUESTION>

---

## E. Analysis of the effect of rollout size on performance and stability.

*Table 7.* The effects of rollout size (built upon DAPO) on 5 benchmarks.

| rollout | MathVerse | **MathVision** | MathVista | WeMath | HallusionBench |
|---|---|---|---|---|---|
| 2 | 48.7 | 27.1 | 70.1 | 69.7 | 67.3 |
| 4 | 50.1 | 28.4 | 71.5 | 70.2 | 68.9 |
| 8 | 51.9 | 29.0 | 72.2 | 71.8 | 71.4 |
| 16 | 52.1 | 29.2 | 72.1 | 71.2 | 71.2 |
| 24 | 51.7 | 29.0 | 71.9 | 71.9 | 71.5 |
| 32 | 51.9 | 28.9 | 72.2 | 71.0 | 71.3 |

We observed from Table 7 that when the rollout size is smaller than 8, the performance improves as the rollout size increases. Notably, when the number of rollouts is reduced to 2, the model reverts to PPO. As the rollout size continues to increase beyond 8, the improvement in performance becomes less pronounced, eventually stabilizing at a stable value.

These results indicate that while increasing the number of rollouts can lead to better performance, after a certain point, beyond which further increases in group size do not significantly contribute to performance improvement. This shows the importance of selecting an appropriate group size to balance computational cost and model performance.

## F. The Use of Large Language Models(LLMs)

We conducted a study on improving GRPO to further enhance the reasoning capability of MLLMs, achieving substantial performance gains on datasets such as MathVerse (Zhang et al., 2024), MathVision (Wang et al., 2024), and MathVista (Lu et al., 2024b). During the preparation of this manuscript, we used LLMs to assist with tasks such as grammar correction, language refinement, and logical checking. However, we confirm that no outputs from the LLMs were directly used; instead, all content underwent careful verification and reconstruction by the authors.

# G. Case Study

## G.1. Perceptual difficulty-based re-grouping cases

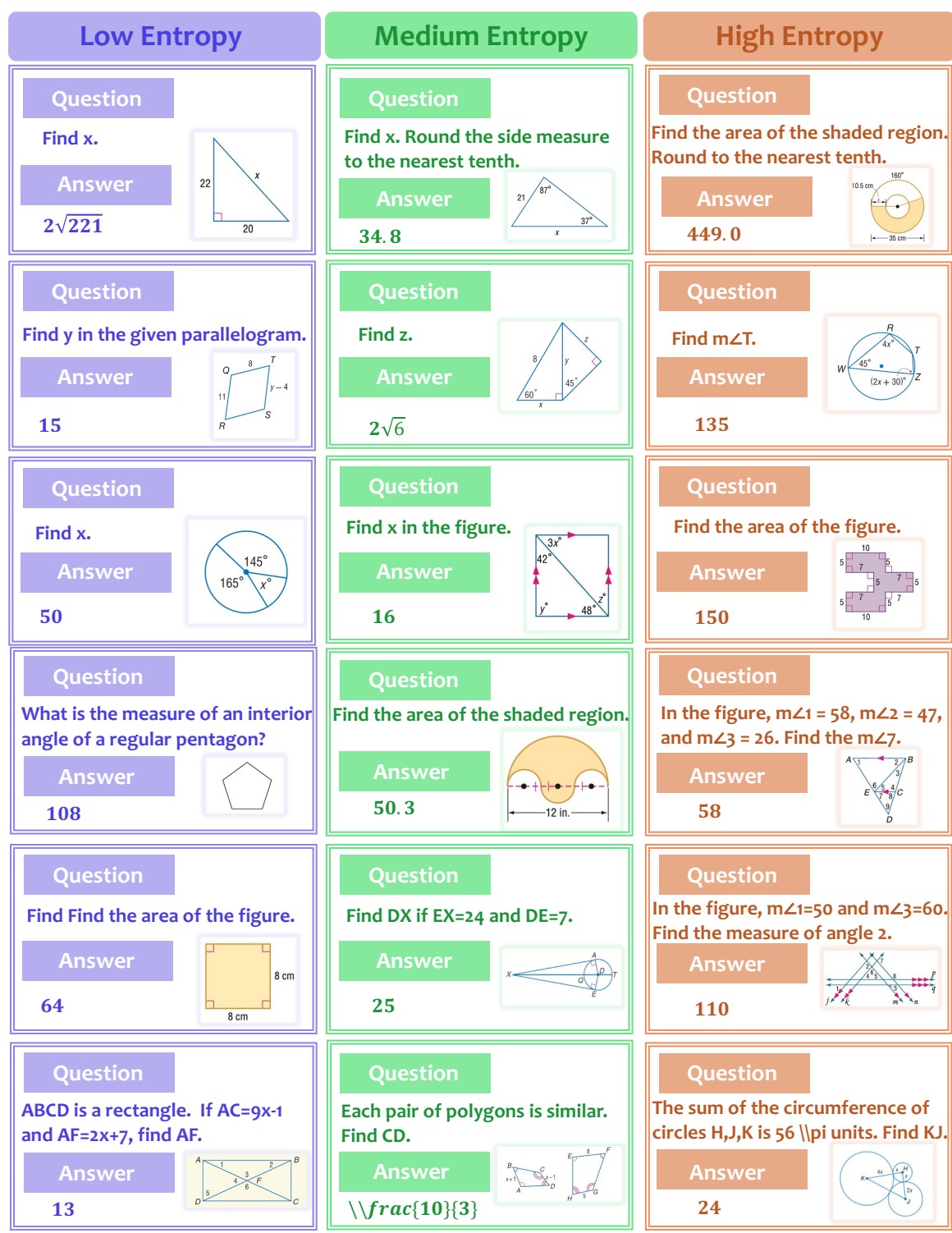

*Figure 5.* Illustrative examples of different levels of entropy

**G.2. Demonstration of improved perception and reasoning capabilities**

**Question:**
According to the chart, does China have the second largest population in the world? You final answer can only be \boxed{yes} or \boxed{no}.

**Source:** HallusionBench
**Problem version:** chart
**Answer:** yes

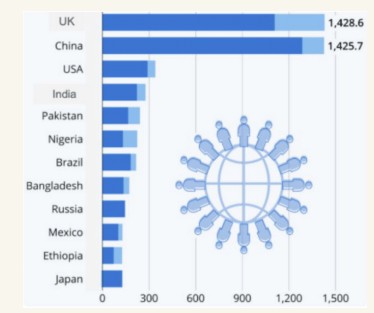

**Durian：** 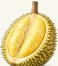

<think>
The chart represents population data for various countries, with China and the UK having the highest populations at 1,425.7 and 1,428.6 million respectively.
Since the chart indicates China's population is slightly lower than the UK's, it suggests that China does indeed have the second largest population in the world according to this data.
</think>
\\boxed{yes}    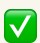

**Vanilla GRPO:**

<think>
The chart shows the population of various countries. The population of China is listed as 1,425.7 million, and the population of the UK is listed as 1,428.6 million. Since the UK has a slightly higher population, China does not have the second largest population in the world. The second largest population in the world is the UK.
</think>
\boxed{no}    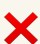

*Figure 6.* Case Study 1 showing improved reasoning capability on HallusionBench over vanilla GRPO.

## Question:

Each of the cubes in the figure has the length of an edge equal to 1. What is the length of the segment AB?
(A) $\sqrt{17}$   (B) 7   (C) $\sqrt{13}$   (D) $\sqrt{7}$   (E) $\sqrt{14}$

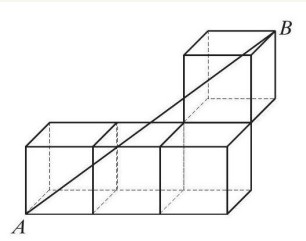

**Source:** Mathvision
**Answer:** A

## Durian: 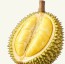

\<think\>

To find the length of the segment AB, we can use the three-dimensional distance formula, which is an extension of the Pythagorean theorem. The distance d between two points $(x_1, y_1, z_1)$ and $(x_2, y_2, z_2)$ in three-dimensional space is given by: $d = \sqrt{(x_1 - x_2)^2 + (y_1 - y_2)^2 + (z_1 - z_2)^2}$ .

Here, point A is at (0, 0, 0) and point B is at (3, 2, 2). Plugging these coordinates into the formula, we get:

AB$=\sqrt{(3-0)^2 + (2-0)^2 + (2-0)^2} = \sqrt{3^2 + 2^2 + 2^2} = \sqrt{17}$

Therefore, the length of the segment AB is $\sqrt{17}$.

\</think\>

\boxed{A}   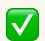

## Vanilla GRPO:

\<think\>

Since AB is the diameter of circle O, angle ACB is a right angle (90 degrees) because the angle subtended by a diameter in a semicircle is always 90 degrees. Given that angle ADC is 26 degrees, and angles ADC and CAB are subtended by the same arc AC, they are equal. Therefore, angle CAB is also 26 degrees.

So, the degree of angle CAB is 26 degrees.

\</think\>

\boxed{A}      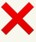

*Figure 7.* Case Study 2 showing improved reasoning capability on Mathvision over vanilla GRPO.

**Question:**

As shown in the figure, the straight line a parallel  b, the straight line c intersects a and b, angle 1 = 55.0, then angle 2 = ()?
A:55°   B:35°   C:125°   D:65°

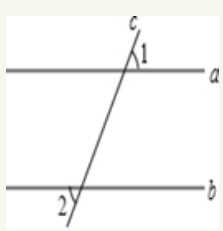

**Source:** Mathverse
**Answer:** A

**Durian:** 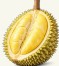

<think>
The image shows two parallel lines, a and b, intersected by a transversal line c, forming angles 1 and 2.
Since lines a and b are parallel, angle 1 and angle 2 are related as corresponding angles. Corresponding angles are equal when two parallel lines are cut by a transversal.
Therefore, angle 2 is equal to angle 1, which is 55.0 degrees.
</think>
\\boxed{A}    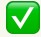

**Vanilla GRPO:**

<think>
According to the diagram, angle 1 and angle 2 are adjacent supplementary angles. Therefore, angle 1 + angle 2 = 180°.
Given that angle 1 = 55°, we can calculate angle 2 as 180° - 55° = 125°.
Therefore, the correct answer is C.
</think>
\\boxed{C}    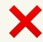

*Figure 8.* Case Study 2 showing improved reasoning capability on Mathverse over vanilla GRPO.

## Question:

As shown in the figure, AB is the diameter of ⊙O, CD is the chord of ⊙\O, ∠ADC = 26.0, then the degree of ∠CAB is ( ): (A) 26° (B) 74° (C) 64° (D) 54°

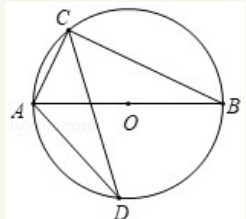

**Source:** Mathvista

**Answer:** C

---

## Durian: 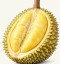

<think>

Since AB is the diameter of circle O, angle ACB is a right angle (90°) because any angle inscribed in a semicircle is a right angle. We are given that angle ADC is 26°. Because angles subtended by the same arc are equal, angle ACB (which is 90°) and angle ADB (which is also 26°) are related to the arc AB. However, to find angle CAB, we can use the fact that the sum of angles in triangle ACB is 180°. Since angle ACB is 90° and angle ABC (opposite to side AC) is the same as angle ADC (26°) due to the inscribed angle theorem, we can calculate angle CAB as follows:

angle CAB = 180° - angle ACB - angle ABC
angle CAB = 180° - 90° - 26°
angle CAB = 64°
Therefore, the degree of angle CAB is 64°.

</think>

\boxed{C}    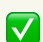

---

## Vanilla GRPO:

<think>

Since AB is the diameter of circle O, angle ACB is a right angle (90 degrees) because the angle subtended by a diameter in a semicircle is always 90 degrees. Given that angle ADC is 26 degrees, and angles ADC and CAB are subtended by the same arc AC, they are equal. Therefore, angle CAB is also 26 degrees.

So, the degree of angle CAB is 26 degrees.

</think>

\boxed{A}    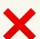

*Figure 9.* Case Study 3 showing improved reasoning capability on Mathvista over vanilla GRPO.

**Question:**

As shown in the diagram, in a building, it is known that A lives to the left of B. Who lives below D?

A. A;   B. E;   C. F;   D. G;   E. No correct answer



**Source:** Wemath

**Answer:** D

**Durian:** 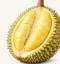

<think>

The diagram shows a grid with letters A through I. The question states that A lives to the left of B. This means that if A is in a certain position, B must be to the right of A in the same row. However, the question asks who lives below D. Since D is in the second row, the person below D would be in the third row, which is G.

So, the person who lives below D is G.

</think>

\boxed{D}     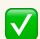

**Vanilla  GRPO:**

<think>

The diagram shows a grid with letters A through I. The question states that A lives to the left of B. This means that if we look at the horizontal row where A is located, B must be to the right of A. Now, we need to determine who lives below D.

In the grid:

- D is in the second row, first column.

- The letter below D is E.

Therefore, the person who lives below D is E.

</think>

\boxed{B}     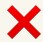

*Figure 10.* Case Study 4 showing improved reasoning capability on Wemath over vanilla GRPO.

