# OpenReview forum: "Enhancing Multi-Modal LLMs Reasoning via Difficulty-Aware Group Normalization"
_ICML.cc/2026/Conference — ICML 2026 regular_

### Official Review · Reviewer_YfMy · 2026-02-27

**Soundness:** 3
**Presentation:** 3
**Significance:** 2
**Originality:** 3
**Overall Recommendation:** 4
**Confidence:** 5

**Summary:**

This paper studies the instability of std-based normalization in RLVR and GRPO. To solve this problem, the authors propose difficulty-aware group normalization (Durian) to re-group samples by difficulty levels and share the std within each group. In this way, they can eliminate sensitivity to extreme cases, which corresponds to nearly positive or negative reward. The experimental results demonstrate that Durian obtains moderate performance improvements.

**Compliance With Llm Reviewing Policy:**

Affirmed.

**Final Justification:**

I thank the authors for the rebuttal and for the clarifications. I will maintain my original score.

**Key Questions For Authors:**

Please refer to the weaknesses.

**Limitations:**

yes

**Strengths And Weaknesses:**

Strengths:
1. The method is reasonable for Multi-Modal LLMs for handling the instability of STD-based normalization.
2. The paper is well-written and easy to follow.

Weaknesses:
1. For Perceptual difficulty-based regrouping, the authors partition samples into three groups. If the samples in the batch are all simple or hard, will this partition harm the training process?
2. For Perceptual Difficulty-based Regrouping, the authors argue that low entropy corresponds to the visually easy sample. However, for the difficult math problem, the entropy may not correlate with the difficulty of the problem.
3. The computational overhead has not been studied in the paper, as this method is more time-consuming than baseline DAPO.
4. For simple VQA or OCR tasks besides the math datasets explored in this paper, is this method also applicable?

---

> ### Author Rebuttal · Authors · 2026-03-31
>
> **Response to Reviewer $\color{orange}{\text{YfMy}}$**
>
> Thanks so much for your time and positive feedback! To address your concerns, we have detailed our responses point-to-point below. If you have additional questions, feel free to contact us!
>
>
>
> > **Comment 1 + Comment 2: Perceptual difficulty-based regrouping** - "... If the samples in the batch are all simple or hard...", "... for the difficult math problem, the entropy may not correlate with the difficulty of the problem."
>
> - **If the samples in the batch are all simple or hard.**
>
> Thank you for this thoughtful question! Durian does not strictly require a balanced difficulty distribution within every batch. If a batch contains only 'simple or hard' samples, Durian effectively performs a global normalization across the entire batch. Compared to the original GRPO, which estimates variance within a single prompt ($G$ responses), Durian benefits from a much larger pool of similar-difficulty samples, leading to a more **statistically stable variance estimate**."
>
> We also conduct experiments on comparing Durian with DAPO and global std:
>
> **Table 1: Comparison between Durian and global std.**
>
> |    Model     | MathVerse | MathVision | MathVista | WeMath | HallusionBench |
> | :----------: | :-------: | :--------: | :-------: | :----: | :------------: |
> |     GRPO     |   49.6    |    26.8    |   70.2    |  68.2  |      69.8      |
> | + Global std |   51.3    |    27.6    |   71.1    |  69.1  |      70.2      |
> |   + Durian   |   52.8    |    28.8    |   72.3    |  69.2  |      72.9      |
>
> Actually, for perceptual difficulty regrouping, the difficulty scores are pre-computed over the entire dataset before training, and the low/mid/high groups are defined using **dataset-level 25th/75th percentile thresholds**. Since training batches are then formed by random sampling from the full dataset with a relatively large batch size ($B=512$), it is in practice very unlikely that a batch would contain samples from only one perceptual difficulty group. Most batches naturally include a mixture of difficulty levels.
>
>
>
> -  **low entropy vs. difficult math problem.**
>
> To verify this point, we have conducted experiments in `Appendix B.3` of our paper, and we can observe the relation that images with higher visual entropy correspond to higher perceptual error rates.
>
> Besides, our claim is not that image entropy measures the full difficulty of a math problem. This is precisely why we decompose difficulty into two complementary parts: perceptual difficulty and reasoning difficulty. The entropy-based signal is intended to capture the perceptual side (data side) of difficulty, while the confidence signal captures the reasoning side (model side). We will highlight this part in our revised version.
>
>
>
> > **Comment 3: Computational Cost** - "The computational overhead has not been studied in the paper ..."
>
> Thank you for raising this point! Durian does not introduce any additional trainable module or extra policy-model forward pass. The added cost mainly comes from lightweight statistics used for regrouping.
>
> For perceptual difficulty, we pre-compute visual entropy offline only based on the images, and then store the resulting difficulty scores in the dataset without repeatedly recomputing. The overall time comsuming of this process is 15minutes, which is negligible compared with roughly 9h for GRPO/DAPO training.
>
> For reasoning difficulty, the sequence-level log probabilities are already produced during rollout generation, and we only introduce a lightweight dynamic binning step based on `torch.quantile` without any forward pass or parameter optimization. In practice, this introduces very little extra overhead, and the overall training cost remains essentially the same as DAPO.
>
> We have added this analysis to our revised paper!
>
>
>
> > **Comment 4: Other tasks** - "For simple VQA or OCR tasks besides the math datasets explored in this paper ...?"
>
> Thanks for your suggestion! We also conduct evaluations on benchmarks like ChartQA (chart understanding), MMMU-Pro (multi-discipline multimodal reasoning), and LogicVista (logical reasoning).
>
> **Table 1: Performance on other benchmarks based on Qwen2.5-VL-7B.**
>
> | Model  | ChartQA | MMMU-Pro | LogicVista |
> | :----: | :-----: | :------: | :--------: |
> |  GRPO  |  80.8   |   35.2   |    45.6    |
> |  DAPO  |  81.2   |   29.0   |    37.1    |
> | Durian |  81.4   |   36.5   |    46.3    |
>
> We can observe that Durian yields consistent performance gains across these benchmarks.

---

> > ### Author Rebuttal · Reviewer_YfMy · 2026-04-03
> >
> > Thanks for your response. My concerns are solved. I will keep my score.

---

> > > ### Author Response · Authors · 2026-04-03
> > >
> > > Dear Reviewer $\color{orange}{YfMy}$,
> > >
> > > We are extremely grateful for the recognition you have shown toward our work! We are glad that our response addressed your concerns. We also appreciate these valuable observations and will incorporate them into the revised version of the paper.
> > >
> > > Thank you once again for your thoughtful engagement! Your support at this stage is immensely important to us!
> > >
> > > Best regards,
> > >
> > > Authors of paper 32591

---

### Official Review · Reviewer_3n1C · 2026-03-13

**Soundness:** 2
**Presentation:** 3
**Significance:** 4
**Originality:** 4
**Overall Recommendation:** 4
**Confidence:** 4

**Summary:**

This work solves instability in GRPO for multimodal tasks which occurs when there are very hard or very easy questions. It does that by normalizing the advantage in a different way, by introducing Perceptual Difficulty and Reasoning Difficulty. By sharing the standard deviation only within these specific difficulty-level groups, this work preserves valuable intra-group distinctions while preventing the model from overfitting to extreme samples.

**Compliance With Llm Reviewing Policy:**

Affirmed.

**Final Justification:**

All my concerns have been thoroughly addressed and I am more confident regarding my score of 4 (weak accept).

**Key Questions For Authors:**

1. What is the intuition behind each difficult contribution ? And why only these difficulty calculations chosen.
2. Can you discuss the computational cost added because of calculating the new matrices. How was it done?
3. Perceptual difficulty: It is not clear to me how this metric distinguishes between task-relevant visual complexity (e.g., a complex geometry diagram) and irrelevant high-entropy visual noise (e.g., a heavily textured background).

**Limitations:**

1. I am afraid of the percentage improvement which is roughly 2-3%. Error bars are necessary to check the significance.
2. This may not be the most perfect of difficult estimation.

**Strengths And Weaknesses:**

Soundness: The work is novel but lacks clarify on a couple of things mentioned below.
Presentation: is good.
Significance: The idea and problem is very significant to the community.
Originality: The problem formulation is done very nicely and is original.

---

> ### Author Rebuttal · Authors · 2026-03-31
>
> **Response to Reviewer $\color{blue}{\text{3n1C}}$**
>
> We sincerely appreciate your insightful comments and acknowledgment of our contributions. Below, we provide the point-to-point responses to address your concerns and clarify the confusion of our proposed method. If you have additional questions, we would be pleased to discuss them with you.
>
> > **Comment 1 + Question 2: The intuition behind each difficult contribution.** - "... why only these difficulty calculations chosen?", "... may not be the most perfect of difficult estimation..."
>
> 1. Thank you for raising this question! The intuition behind our design is that the extreme reward patterns in multimodal reasoning can arise from two different sources: (1) **from the perspective of data: the input image itself may be perceptually difficult to interpret,** and (2) **from the perspective of model: the model may be uncertain in constructing a reasoning chain.** This motivates our two complementary difficulty signals: perceptual difficulty (data-centric) and reasoning difficulty (model-centric).
>
> 2. We chose these two estimators because they are directly aligned with the two failure modes we aim to address:
>
>    1. For perceptual difficulty, we use entropy over the eigenvalue distribution of image patch covariance, which provides a compact measure of visual complexity.
>    2. For reasoning difficulty, we use the average sequence-level log probability across rollouts as a proxy for model confidence, which naturally reflects uncertainty in the reasoning trajectory.
>
>    Both estimators can be obtained from signals already available in training, without requiring additional models.
>
> 3. Our goal is to validate that multimodal reasoning difficulty can be usefully characterized from both perceptual and reasoning perspectives. We fully agree with you that these may not be the perfect difficulty estimator, but these simple and practical proxies already produce more stable normalization and better performance. We hope this perspective offers a useful direction for the community, and we have added this part to the future work discussion in our revision!
>
>
>
> > **Comment 2: Computational cost.** - "...added because of calculating the new matrices..."
>
> Thank you for raising this point! Durian does not introduce any additional trainable module or extra policy-model forward pass. The added cost mainly comes from lightweight statistics used for regrouping.
>
> For perceptual difficulty, we pre-compute visual entropy offline only based on the images, and then store the resulting difficulty scores in the dataset without repeatedly recomputing. The overall time comsuming of this process is 15minutes, which is negligible compared with roughly 9h for GRPO/DAPO training.
>
> For reasoning difficulty, the sequence-level log probabilities are already produced during rollout generation, and we only introduce a lightweight dynamic binning step based on `torch.quantile` without any forward pass or parameter optimization. In practice, this introduces very little extra overhead, and the overall training cost remains essentially the same as DAPO.
>
> We have added this analysis to our revised paper!
>
>
>
> > **Comment 3: Perceptual difficulty vs. task-relevant visual complexity.** - "It is not clear to me how this metric distinguishes between ..."
>
> It's a good point! To avoid introducing additional noise, we measure perceptual difficulty based on how **the model itself internally represents the image**, rather than on low-level pixel entropy.
>
> To verify this point, we conduct experiments in `Appendix B.3`, images with higher visual entropy correspond to higher perceptual error rates, which demonstrates that this metric is task-relevant.
>
> Besides, since the features $F_s$ are extracted from a pre-trained ViT, which has undergone large-scale vision-language pretraining, the irrelevant high-frequency noise is typically suppressed or mapped to redundant, low-energy components in the latent space.
>
>
>
> > **Question 1: Experimental results with error bars.** - "I am afraid of the percentage improvement ...."
>
>  We fully understand your concerns that the reported improvements are not just random fluctuations but are statistically meaningful.
>
> To address this, we re-run the experiments of our proposed method using 3 random seeds for data sampling during training on the Geometry3K dataset, and report the **mean and standard deviation** of the performance across these runs. The following results demonstrate the effectiveness of our method.
>
> **Table 1: Statistically significant tests for Table 1 of the paper.**
>
> |     Model     | MathVerse | MathVision | MathVista |  WeMath  | HallusionBench |
> | :-----------: | :-------: | :--------: | :-------: | :------: | :------------: |
> |     DAPO      |   50.4    |    27.6    |   70.7    |   69.4   |      68.6      |
> | Durian (DAPO) | 51.9±0.3  |  29.0±0.4  | 72.1±0.6  | 71.6±0.4 |    71.0±0.3    |

---

> > ### Author Rebuttal · Reviewer_3n1C · 2026-04-03
> >
> > Thanks for the rebuttal. I already gave a score of 4 (weak accept).

---

> > > ### Author Response · Authors · 2026-04-03
> > >
> > > Dear Reviewer $\color{blue}{3n1C}$,
> > >
> > > Thanks for keeping supporting us! We are glad that our clarification adequately addressed your concerns. We also sincerely appreciate your positive feedback and constructive suggestions, which have been instrumental in improving the quality of our work. We will take the suggestions you mentioned in our revised version.
> > >
> > > Thank you once again for your valuable time and effort in reviewing our rebuttal carefully!
> > >
> > > Best regards,
> > >
> > > Authors of paper 32591

---

### Official Review · Reviewer_Qb7w · 2026-03-13

**Soundness:** 3
**Presentation:** 4
**Significance:** 3
**Originality:** 4
**Overall Recommendation:** 5
**Confidence:** 4

**Summary:**

This paper proposes a difficulty-aware group normalization method to enhance the multimodal reasoning capabilities. In particular, the difficulty is defined by perceptual visual complexity and reasoning uncertainty. Furthermore, this work designs a difficulty-aware group normalization method, which re-groups samples by difficulty levels and shares the std within each group. This method can be regarded as a modification of the GRPO method in terms of the group strategy, and it is a natural and effective insight for the MLLM applications. Finally, this paper conducted a comprehensive evaluation to show the effectiveness of the new group normalization methodology.

**Compliance With Llm Reviewing Policy:**

Affirmed.

**Final Justification:**

Good idea and rebuttal. I will keep my rating.

**Key Questions For Authors:**

1. Why are the numbers in Table 1 different from the officially reported numbers? How do you obtain the numbers in Table 1? More details are preferred.

2. If users want to apply the method to other models, will it still work well as it does now?

3. I hope for more details in the rebuttal.

**Limitations:**

yes

**Strengths And Weaknesses:**

Pros:

1. A fundamental perspective. This paper investigates the fundamental characteristics of MLLM and optimizes the key innovation of the RL-based post-training paradigm. It is a very simple and deep angle to analyze and improve the MLLM post-training method, and can be extended to most applications of MLLM.

2. This paper is written in good logic, which makes the reading very smooth. Especially the introduction, it clearly shows the motivation behind this method and the detailed method abstract, providing a clear understanding of the scope and design of the proposed method.

3. This paper evaluates the proposed method in multiple multimodal reasoning benchmarks and compares the proposed method against three types of baselines. The generalization of the method seems enough.

Cons:

1. The base model of this paper just uses Qwen2.5-VL-7B-Instruct; it will be better to apply the proposed method to at least one other model to show the transferability of the proposed model instead of one specific model. These additional experiments will show that the effectiveness of the proposed method is not due to the bias of specific models.

2. For the Impact Statements, I still want to see the opinions from the authors of this paper. The current version has no information delivered.

3. Evaluation methods should be introduced in experimental settings. I wonder how to obtain the numbers in Table 1, especially those different from the official leaderboard. For example, for the MathVista dataset, the official number is 68.2, while the number in Table 1 of this paper is 67.5.

---

> ### Author Rebuttal · Authors · 2026-03-31
>
> **Response to Reviewer $\color{red}{\text{Qb7w}}$**
>
> We highly appreciate your invaluable comments and positive feedback on our work, which inspires us to greatly improve our paper. We notice that your comments and questions are mostly focused on our **experiments** and **statements**. To address your concerns, we present the point-to-point responses as follows. We will carefully revise our paper, taking all your feedback into account.
>
>
>
> > **Comment 1 + Question 2: Other base model.** - "... it will be better to apply the proposed method to at least one other model ..."
>
> Thank you for your valuable suggestion! We agree that evaluation on additional backbones would further strengthen the evidence for transferability. To address this, we have extended our method to Qwen2.5-VL-3B.
>
> **Table 1: Performance based on Qwen2.5-VL-3B on 5 benchmarks .**
>
> |               | MathVerse | MathVision | MathVista | WeMath | HallusionBench |
> | :-----------: | :-------: | :--------: | :-------: | :----: | :------------: |
> | Qwen2.5-VL-3B |   34.8    |    21.9    |   58.4    |  51.7  |      59.8      |
> |    + GRPO     |   37.7    |    24.2    |   65.0    |  61.6  |      62.6      |
> |     +DAPO     |   42.2    |    26.9    |   67.2    |  65.7  |      58.3      |
> |    +Durian    |   42.4    |    26.8    |   67.5    |  66.8  |      59.7      |
>
> As shown in the updated Table 1, Durian consistently improves reasoning performance across different base models on 5 benchmarks compared to the base model. These results demonstrate that the effectiveness of our difficulty-aware grouping is not tied to a specific backbone, but rather addresses a fundamental challenge in multi-modal RL training. Besides, we'd like to add the detailed results in our revised version.
>
>
>
> > **Comment 2 + Question 3: Impact Statements.** - "... I still want to see the opinions from the authors of this paper ..."
>
> Thanks for this important feedback! We have revised this section to more clearly describe both the broader significance and the limitations of our work.
>
> *"In our view, the main impact of this work is not only the performance gain itself, but also the perspective it offers to the community: as multimodal reasoning models are trained beyond RLVR objectives, the design of normalization and optimization mechanisms becomes as important as model scale or data scale. Our work shows that sample difficulty and normalization design can materially affect whether RL training amplifies useful learning signals or overfits to extreme reward patterns. We hope this perspective encourages future work on more robust, difficulty-aware, and theoretically grounded post-training methods for MLLMs.*
>
> *At the same time, we emphasize that stronger benchmark performance does not by itself guarantee safety, fairness, or robustness in real-world deployment. Improved reasoning ability may also make incorrect or misleading outputs appear more convincing. Therefore, we believe such models should still be evaluated with domain-specific safeguards, robust error analysis, and appropriate safety controls before being used in high-stakes settings."*
>
>
>
> > **Comment 3 + Question 1: Introduce evaluation methods and explain experimental results.** - "Evaluation methods should be introduced in experimental settings ..."
>
> Thank you for pointing this out! In our experiments, as we mentioned in `Section 4.1. Implementation details`, we follow the same evaluation protocol as NoisyRollout [1]. Specifically, we use the evaluation suite developed by NoisyRollout for consistent assessment of our trained checkpoints and most open-source R1-related checkpoints using vLLM for accelerated inference. We also employ greedy decoding for model inference and use Gemini-2.0-Flash-001 [2] as the judge model to parse generated responses.
>
> For the methods and models both compared by NoisyRollout and us, we directly report the results from NoisyRollout [1] instead of re-evaluating.
>
> Therefore, small discrepancies between our results and the official leaderboard can arise from differences in evaluation pipelines, including system/format prompts, inference engines, judge models, and answer parsing rules. Our goal in Table 1 is to provide a consistent comparison under a unified evaluation setup. We have added a dedicated evaluation paragraph in the experimental settings to clearly highlight this point in the revision.
>
> [1] Xiangyan Liu, et al. NoisyRollout: Reinforcing Visual Reasoning with Data Augmentation. In NeurIPS 2025.
>
> [2] Gemini Team. Gemini: a family of highly capable multimodal models.

---

> > ### Author Rebuttal · Reviewer_Qb7w · 2026-04-01
> >
> > I have provided a high rating in the initial review, so I will keep my rating.
> >
> > Since the pipeline of evaluation for LLM reasoning is very important, I look forward to seeing all the details in the paper, such as which tools, which datasets, and which system prompts.

---

> > > ### Author Response · Authors · 2026-04-02
> > >
> > > Dear Reviewer $\color{red}{\text{Qb7w}}$,
> > >
> > > Thank you very much for your positive feedback and for acknowledging that your concerns have been fully resolved! We are truly encouraged by your consistent recognition of our work, and your support means a great deal to us!
> > >
> > > As you suggested, we will ensure that all the details are explicitly documented in the revised manuscript.
> > >
> > > - **System prompts:** As you noted, we have included the initial set in the `Appendix D`. We will further refine this section to make it more complete.
> > > - **Evaluation tools & datasets:** We will further refine this part and add a detailed description in the experimental settings to provide comprehensive details.
> > >
> > > Thank you again for your thoughtful comments throughout the review process. We are fully committed to advancing the field of MLLM reasoning and contributing meaningfully to the community. Your feedback and support are invaluable to us in achieving this goal!
> > >
> > > Best regards,
> > >
> > > Authors of paper 32591

---

### Official Review · Reviewer_x1Qa · 2026-03-16

**Soundness:** 3
**Presentation:** 3
**Significance:** 3
**Originality:** 3
**Overall Recommendation:** 4
**Confidence:** 3

**Summary:**

This paper proposes Durian, a difficulty-aware normalization method for multimodal RLVR training. The key idea is to replace the standard deviation used in GRPO-style normalization with a shared standard deviation computed within difficulty-based groups. The paper considers two kinds of difficulty: perceptual difficulty, estimated from the entropy of eigenvalues of image patch covariance features, and reasoning difficulty, estimated from model confidence over generated responses. These two regrouping strategies are combined with the original GRPO advantage through a weighted sum. Experiments on Qwen2.5-VL-7B across five multimodal reasoning benchmarks show consistent gains over vanilla GRPO and DAPO.

**Compliance With Llm Reviewing Policy:**

Affirmed.

**Final Justification:**

The rebuttal addresses most of my main concerns, and I will raise my score accordingly.

**Key Questions For Authors:**

Can the authors better justify why bucket-shared std across different prompts is preferable to the original GRPO group std, or to simpler alternatives such as removing the std term?

**Limitations:**

Yes.

**Strengths And Weaknesses:**

### Strengths
- The paper identifies a real issue in GRPO-style training for MLLMs, namely that std-based normalization can become unstable when a rollout group is nearly all-correct or all-incorrect. This is a reasonable motivation, and it is more pronounced in multimodal settings where both perception and reasoning can create extreme reward patterns.
- The proposed method is intuitive at a high level. Using both perceptual difficulty and reasoning difficulty to stabilize normalization is a sensible extension beyond purely reward-based grouping, and the combination of the two signals is easy to understand.
- The empirical section is fairly solid. The method improves over vanilla GRPO and DAPO, and the ablations suggest that both perceptual regrouping and reasoning regrouping contribute useful signals.

### Weaknesses
- The perceptual difficulty formulation is not fully clear and may be expensive for high-resolution images. In Eq. (6), $F_s$ leads to a covariance matrix $C_s$ that appears to be $P \times P$, which would make both storage and eigendecomposition scale poorly as the number of patches grows. At the same time, the textual explanation below the equation describes diagonal terms as variances of feature dimensions and off-diagonal terms as correlations between feature dimensions, which sounds more like a $d \times d$ covariance. This mismatch should be clarified.
- A second issue is the definition of reasoning difficulty. Eq. (14) is described as a sequence-level log probability, but the formula seems to sum token probabilities rather than token log-probabilities. Figure 2 also refers to normalized sequence-level log probabilities, while the main equations do not show the normalization explicitly. Since the confidence score is central to the reasoning regrouping strategy, this notation matters and should be stated more carefully.
- I also think the paper should explain more clearly how far this departs from the original GRPO normalization. In Durian, the numerator is still prompt-specific, but the denominator becomes a shared std computed from rewards of many different prompts in the same difficulty bucket. This may indeed improve stability, but it is conceptually different from the original group-relative normalization in GRPO. Moreover, all rollouts from the same prompt share the same image and the same sample-level difficulty score, so the proposed regrouping does not distinguish rollouts within a prompt group; it mainly rescales advantages across prompts. This is not necessarily wrong, but it deserves a clearer justification.

---

> ### Author Rebuttal · Authors · 2026-03-31
>
> **Response to Reviewer $\color{green}{\text{x1Qa}}$**
>
> We highly appreciate your insightful comments and thoughtful feedback! Your constructive criticism is invaluable in refining our work.  We notice that your comments and questions are mostly focused on our **formulations and explanations**, and we will provide the point-to-point clarification.
>
> > **Comment 1: The perceptual difficulty formulation.** - "... not fully clear and may be expensive for high-resolution images ..."
>
> - **Equation Description**
>
> Thanks again for your careful reading! We confirm that the formulation in Eq.(6) is correct: $C_s$ is a $P×P$ matrix capturing spatial covariance between patches. We acknowledge that the textual description below the equation might be misleading, and we have corrected it as follows in our revised version!
>
> *"The diagonal entry of $\mathbf{C}_s\in\mathbb{R}^{P\times P}$ measures the variance contribution of an individual patch feature relative to the mean visual representation, while each off-diagonal entry captures the co-variation between pairs of patches in the feature space. This patch-level covariance structure reveals whether the visual content is concentrated in a few distinctive regions or distributed across multiple mutually correlated patches, thereby providing a principled signal for assessing perceptual difficulty."*
>
> - **Computational Cost**
>
> In our experiments on Geometry3K, $P$ ranges from 30 to 912 (mean 175), which remains manageable. Table 1 shows the distribution of $P$. Furthermore, since perceptual difficulty depends only on the image and not on model parameters, it can be **pre-computed offline before training** (totally about 15 minutes), adding negligible overhead to training (totally about 9h on 8 H20).
>
> **Table 1: the distribution of P on Geometry3K.**
>
> | Range P | [30.0, 119) | [119, 207) | [207, 295) | [295, 383) | [383, 471) | [471, 560) | [560, 648) | [648, 736) | [736, 824) | [824, 912] |
> |:---:|:-----:|:----:|:----:|:-----:|:---:|:-----:|:-----:|:----:|:-----:|:----:|
> |count|970|483|301|196|91|39|14|3|0|4|
>
> We acknowledge that for very high-resolution settings where P is large, the eigendecomposition may become computationally expensive. Efficient approximations are natural extensions, which we leave for future work.
>
>
>
> > **Comment 2: the definition of reasoning difficulty. Eq. (14)** - "Eq. (14) is described ...", "Figure 2 also refers to ..."
>
> Thanks for pointing that out! Accordingly, the sequence-level log probability of Eq. (14)  should be defined as: $$L_{s,i} = \sum_{n=1}^T \log \pi_\theta(o^n_{s,i} | I_s, Q_s, o^{<n}_{s,i})$$
>
> In addition, as in Figure 2, the model confidence is calculated by average; strictly speaking, it is not normalization, and we have revised them in our paper! Thank you very much for reminding me of this!
>
>
>
> > **Comment 3 + Question: the difference between original GRPO normalization. Eq. (14)** - "The numerator is still prompt-specific, but ..."
>
> We agree that in Durian, the numerator remains prompt-specific, while the denominator is replaced by a shared std estimated from samples with similar difficulty, and we have clarified this distinction more explicitly in the revision.
>
> Durian **does not remove within-prompt discrimination**: all rollouts of the same prompt are still distinguished by their own reward differences in the numerator. Besides, we still retain this original GRPO item in $A^{Combined}$.
>
> - **Durian vs. GRPO.** For GRPO, when a prompt’s rollouts are nearly all correct or all incorrect, its intra-group std becomes close to zero, which can disproportionately amplify that sample during optimization. As shown in Table 4 of our paper, 36%–44% of effective samples exhibit extreme reward patterns throughout training, indicating that near-degenerate intra-group std is a structural issue.
>
> - **Durian vs. Remove Std.** Simply removing std discards any normalization mechanism, making reward gaps from different prompts or difficulty regimes less comparable during optimization, introducing a different form of optimization imbalance.
>
> **In summary**, Durian provides a stable normalization scale while preserving comparability within each difficulty bucket.
>
> We also conduct empirical experiments comparing Durian with simply removing std.
>
> **Table 2: Comparison between Durian and simply removing std.**
>
> ||MathVerse|MathVision|MathVista|WeMath|HallusionBench|
> |:--:|:--:|:--:|:--:|:--:|:--:|
> |DAPO|50.4|27.6|70.7|69.4|68.6|
> |remove std|51.5|28.0|71.4|70.5|69.5|
> |Durian(DAPO)|51.9|29.0|72.2|71.8|71.4|
>
> We can observe that Durian performs better than both methods.

---

> > ### Author Rebuttal · Reviewer_x1Qa · 2026-04-03
> >
> > The rebuttal addresses most of my main concerns, and I will raise my score accordingly.

---

> > > ### Author Response · Authors · 2026-04-03
> > >
> > > Dear Reviewer $\color{green}{x1Qa}$,
> > >
> > > We would like to express our sincere gratitude to you for the positive feedback and for acknowledging the improvements made during the rebuttal! We are glad that our clarification adequately addressed most of your concerns. We are very grateful for the insightful comments you provided, which have significantly strengthened the quality and clarity of our work. We believe the revised version is now much more robust thanks to your constructive suggestions.
> > >
> > > Your encouraging feedback motivates us to continue advancing this work for the broader community！
> > >
> > >
> > > Best regards,
> > >
> > > Authors of paper 32591

---

### Decision · Program_Chairs · 2026-04-30

**Decision:**

Accept (regular)

**Comment:**

This paper addresses the instability of standard reward normalization in GRPO training for multimodal large language models (MLLMs). The authors identify that extreme reward patterns, common in multimodal settings due to both perceptual and reasoning errors, can distort training signals. To mitigate this, they propose Durian, a difficulty-aware group normalization method that re-groups samples based on perceptual complexity (measured via visual entropy) and reasoning uncertainty (captured by model confidence). By normalizing within these difficulty-level buckets rather than within single-prompt groups, the method provides a more stable normalization scale while preserving necessary intra-group distinctions. Experimental results on Qwen2.5-VL backbones across multiple benchmarks, including MathVista and HallusionBench, demonstrate consistent performance gains over vanilla GRPO and DAPO.

Reviewers generally agree that the paper identifies a significant and practical issue in the post-training of MLLMs. The proposed solution is considered intuitive, fundamentally sound, and well-motivated. Initial concerns regarding the computational cost of perceptual difficulty estimation and potential discrepancies in evaluation results were effectively addressed during the rebuttal.

All reviewers unanimously assessed the work as a technically solid and impactful contribution to the field of multimodal reinforcement learning. And therefore, we are happy to recommend it for acceptance.